# Mixing Job Training with Nature-Based Therapy Shows Promise for Increasing Labor Market Affiliation among Newly Arrived Refugees: Results from a Danish Case Series Study

**DOI:** 10.3390/ijerph19084850

**Published:** 2022-04-16

**Authors:** Sigurd Wiingaard Uldall, Dorthe Varning Poulsen, Sasja Iza Christensen, Lotta Wilson, Jessica Carlsson

**Affiliations:** 1Competence Centre for Transcultural Psychiatry (CTP), Mental Health Centre, 2750 Ballerup, Denmark; jessica.carlsson.lohmann@regionh.dk; 2Department of Geosciences and Natural Resource Management, University of Copenhagen, 1198 København, Denmark; dvp@ign.ku.dk; 3New Roots, Hallingelille Økosamfund, 4100 Ringsted, Denmark; sasja@siza.dk (S.I.C.); lottawilson@hotmail.com (L.W.); 4Institute of Clinical Medicine, Faculty of Health and Medical Sciences, University of Copenhagen, 2200 København, Denmark

**Keywords:** nature-based therapy, horticulture, employment, job training, refugees, PTSD

## Abstract

The unemployment rate among newly arrived refugees in European countries is high and many experience mental health problems. This has negative consequences on integration and mental well-being. In this case series study we investigated the effect of a 30-week program that included horticulture activities, nature-based therapy, and traditional job training on job market affiliation and mental well-being. Fifty-two refugees met initial screening criteria and twenty-eight met all inclusion criteria and were enrolled. The program took place in a small community and consisted of informal therapeutic conversations, exercises aimed at reducing psychological stress, increasing mental awareness and physical wellbeing. At the end of the program traditional job market activities were led by social workers. Provisionary psychiatric interviews showed that at baseline 79% met criteria for either an anxiety, depression, or PTSD diagnosis. After the program, statistical analyses revealed an increase in the one-year incidence of job market affiliation (n = 28) and an increase in mental health according to two of four questionnaire measures (n_range_ = 15–16). The results strengthen the hypothesis that horticulture and nature-based therapy can help refugees enter the job market. However, the small sample size emphasizes the need for methodologically stronger studies to corroborate these preliminary findings.

## 1. Introduction

Ten percent of the world’s 26 million refugees live in European countries [1] and the increasing inflow in 2013–2016 has caused political and economic challenges to most Western countries [2]. Successful refugee integration can be halted by low employment rates [3,4] and its bearings on other integration domains such as language skills and citizenship [5]. Despite recent years’ attempts to facilitate labor market integration [6], Denmark, similar to other European countries [7], continue to report high refugee unemployment rates, such as 64% and 49% for refugees with residence between 0–3 years and 3–6 years in 2020, respectively [8]. While the explanation for these rates are multifactorial, recent research has shown that unemployment among refugees is interrelated with cultural and language differences [9,10] and psychological stress [11]. Against the backdrop of other significant stressors such as pre-migratory traumatic experiences (e.g., war, torture, famine) and stressors related to the migration, unemployment is moreover a potential risk factor for mental disorders among refugees [12,13], including posttraumatic stress disorder (PTSD) and depression [14]. While research has demonstrated the detrimental impact of both unemployment and mental distress among refugees there is a lack of studies investigating possible mitigating interventions.

Meanwhile, an increasing body of research shows that nature-based therapy can have a positive impact on stress [15,16,17] and mental disorders, including PTSD [18,19] and anxiety [20]. Nature-based therapy covers a range of activities that pivot on experiences in nature and facilitates treatment processes. One widespread form of delivering nature-based therapy is horticulture, of which many activities have been adopted from occupational therapy [21,22]. The dominating theory corroborating the therapeutic aspect of nature is attention restoration theory (ATR) developed by Kaplan [23]. ART suggests that nature has a restorative effect on cognitively demanding forms of attention that we employ voluntarily and with effort, such as when the attention goes ‘against the grain’. Nature’s restorative effect is assumed to be mediated by mental states of effortless fascination and associated positive emotional and existential states, which have been substantiated by environmental studies on stress-related conditions linking the experience of drifting clouds [24], calmness, and sense of freedom [25] with improved mental health. 

Generally, nature-based therapy is comprised of activities and experiences in nature that aim at integrating psychological and physical perspectives [26]. The interventions often strive to comply with theories of salutogenesis, as developed by Aaron Antonovsky [27], and emphasize an individual’s resources rather than, for example, focusing on symptoms of psychological stress. Furthermore, the therapeutic interventions are designed to provide a high level of manageability, meaning, and comprehensibility which allows for a sense of coherence (SOC) to be experienced. In a nature-based and horticultural therapeutic intervention program, the specific exercises consider the psychological and physical characteristics of the target group and allow for a gradual intensification such that they are experienced first as supportive and later more challenging. 

To meet the challenges of high unemployment rates and prevalence of mental distress in refugees, the present study investigated the outcomes of a 30-week program consisting of nature-based therapy and activities directly focused on preparing for a return to the labor market [28,29]. Residing on theories of salutogenesis, the program strived to enhance communication skills, increase a sense of community, and bridge cultural differences. We hypothesized that the intervention would increase affiliation to the labor market by increasing employment rates and improving mental well-being.

## 2. Materials and Methods

### 2.1. Study Design

The case series study was conducted at Hallingelille Ecovillage in Region Sjaelland, Denmark (https://www.hallingelille.dk/ (accessed on 15 April 2022, in Danish)). The program was directed by two therapists, each with extensive clinical experience in psychotraumatology from working with refugees from the Danish Red Cross asylum department. Moreover, the therapists were trained in ecotherapy and psycho organic analysis at teaching facilities whose programs rest on theoretical work by, among others, Linda Buzzell [30] and Bessel van der Kolk [31]. Volunteers from the ecovillage took part in the majority of daily activities. 

Hallingelille Ecovillage houses approximately 50 adults and 30 children and is characterized by self-sufficient ecological living. It is situated in green surroundings and covers approximately 60 acres with vegetable farms and domesticated animals. Part of the village is designed to facilitate nature-based therapy and includes a bonfire cottage, a greenhouse, and an outdoor kitchen. 

The program was arranged in collaboration with five job centers bordering the ecovillage. A job center is a municipal body that coordinates job programs and participation is typically a requirement for receiving public unemployment benefits. The present program constituted an employment program activity and was offered to eligible citizens (see below). The job centers emphasized for all potential participants that to reject would have no negative consequences, e.g., withdrawal of employment benefits. While enrolled in the program participants were exempt from other activities arranged by the job center. All invited job centers agreed to participate.

### 2.2. The Design of a Nature-Based Therapy Program 

The foundation of the program was nature-based therapy developed by researchers at the University of Copenhagen, Denmark. The program was designed to specifically enhance psychological and physical factors of relevance for the labor market and was divided into three consecutive periods, each lasting ten weeks with increasing workload (see Figure 1). All participants underwent the same exercises, but the intensity was adjusted according to the psychological and physical resources of the individual. Through the program, participants were continuously encouraged to take on activities with increasing physical and psychological demands and to select activities out of curiosity and skills (see below). Participants worked either alone or in groups of 2–4, and between 5 min (e.g., relaxation techniques, see Table 1) to several hours (e.g., horticultural activities), depending on the activity. Interpreters were available during (1) introduction to the program, (2) introduction to individual exercises, (3) therapy sessions, (4) program completion, and (5) psychoeducation. Language barriers during additional instruction and communication were bridged using gesticulations and drawings. 

Throughout the program the two therapists would seek out participants and conduct informal and non-directive conversations on topics such as general well-being, family, culture, how they thrive in the project and their mental state. 

In Period 1, the aim was first to create a safe attachment to the village. This was achieved by helping participants discover and establish a safe and private physical space and by facilitating strong emotional relationships between participants, typically in groups of 2–5. Participants were presented to exercises designed to reduce psychological stress and increase mental awareness and physical wellbeing (see Table 1). Participants were encouraged to practice these exercises on their own both in and outside the village. Furthermore, participants were instructed in light (e.g., sowing, planting, and weeding) and more physically demanding (e.g., digging, stacking firewood, and weeding in vegetable fields) horticultural activities and gardening. 

The overall aims of Period 2 were to enhance communication skills and a sense of community. Participants were psychoeducated on psychological and bodily reactions to psychological traumas and the possible positive impact of nature on mental health. In this vein, participants were encouraged to pay attention to bodily expressions of mental states and their possible interactions with different activities throughout the program. To increase sense of community, participants were instructed to undertake horticultural activities in collaboration with co-participants, including tasks for the benefit of all participants, such as preparations of bonfire for cooking. Additionally, as a group the participants went hiking in the woods, were instructed in caring for domesticated animals in the village (e.g., sheep), and participated in regular communal dining. To begin the process of clarifying personal potential, interests, and goals, participants commenced working with their personal life story using creative arts, such as painting, drawings, and clay. This process was facilitated by the therapists in exploratory conversations using open-ended questions. For one day the closest family members of the participants were invited to the village and to participate in the program. 

The third period aimed more specifically at strengthening work competence. Participants continued to perform daily activities in the village as detailed above but were now strongly encouraged to choose based on curiosity and skills, and not only mental and physical resources. The type of activity was extended to include work at a wood studio, harvesting crops, construction and repairing assignments (e.g., kitchen machines, changing oil, building bird boxes or garden benches), caring for hives and animals at the village, sale of local groceries and customer contact in the village store, and participation in workdays arranged for all residents in the village. Participants were encouraged to use the art and craft activities to express emotions, e.g., related to previous traumatic experiences, and to verbalize them in follow-up conversations with the therapists. To simulate ordinary work conditions and employ cognitive functions typically required on the labor market such as working memory, problem solving, and concentration, participants were assisted in designing personal projects with an evaluable result. Concurrent to these activities, participants held meetings with social workers who had expertise in helping the unemployed into the labor market. Based on the uncovered personal resources and goals, the social worker helped facilitate relevant internships, education, and employment. Expeditions to companies, e.g., farms, forestry, agriculture, construction, and educational institutions were arranged and in-depths conversations with the therapists concerning work expectations, opportunities, and dreams, including cultural aspects, were carried out. As in Period 2, close family were invited to the village and to participate in the program for one day.

### 2.3. Participants

The program was carried out from April–November in both 2017 and 2018. In both years participants were recruited in the preceding months at five job centers bordering the intervention site. Each job center screened their clients using the following criteria: (1) a maximum of 5 years residence in Denmark on the ground of being a refugee; (2) a record of failing Danish courses and hence a minimal Danish language level; (3) a record of being unable to complete employment program activity because of physical complaints (e.g., pain) or mental health complaints (e.g., symptoms of anxiety or depression). To exclude participants with a psychotic condition or alcohol/substance abuse, eligible participants were interviewed with the short full structured Mini-International Neuropsychiatric Interview, version 5, (MINI-5) [32]. The MINI-5 provides provisional psychiatric diagnoses according to the Diagnostic and Statistical Manual IV (DSV-4, Association, 1994) [33]. If the interview resulted in a tentative diagnosis (see Table 2), the participant was informed and encouraged to contact his/her family doctor for further diagnostic clarification. Due to the nature of the study design the number of participants included were not driven by a power calculation but by feasibility considerations. We aimed at including a total of 40 participants (20 each year). Transportation to and from the site was arranged by volunteers and there were no economic expenditures for the participants. 

Overall, 72 citizens were deemed eligible by caseworkers at the job centers based on their record profile, of which 65 agreed to meet with a research assistant for additional screening for eligibility. Overall, 52 participants met inclusion criteria and agreed to be interviewed with the MINI-5. The interview led to the exclusion of 2 participants due to suspicion of a psychotic condition and 5 participants due to the presence of substance abuse. Of the remaining 45 participants, 7 participants withdrew informed consent due to change-of-mind before study initiation, leaving 38 to be enrolled at start-up. After program initiation, 10 participants (6 from group 1) dropped out during the first period of the program; 1 was physically too weak to participate, 1 had a change of mind when he received pension shortly after commencing the program, 1 unexpectedly left for his home country, 4 had worsening of chronic pain, and 2 had severe worsening of mental health issues. Additionally, 1 participant was excluded due to signs of alcohol abuse. Of the remaining 28 participants, 15 completed the program in 2017 (group 1) and 13 in 2018 (group 2). The quantitative variables from the available questionnaires at baseline for the 10 dropouts in Period 1 fell within one standard deviation from the mean of the completers. Demographic characteristics of the 28 completers are shown in Table 2. Participants were from Syria, Congo, Iran, Iraq, Afghanistan, Kurdistan, Eritrea, Chechnya, and Russia and were fluent in either Arabic, Farsi, Chechen, Tigrinya, Kurmanji, or English.

### 2.4. Outcome Measures 

The primary outcome was labor market affiliation defined as internship, educational training, vocational training, or employment. A one-year incidence of failed, initiated, and completed activities were recorded at baseline and one-year post-program completion. The reported activities were confirmed by the job center. For each participant we constructed two affiliation scores (pre-program initiation and post-program completion) with each activity counting equally and dichotomized the score into an affiliation categorical variable for each participant (Yes/No). The categorical variable was used to test the hypothesis that affiliation to labor market would increase following the program. 

As secondary outcomes we assessed symptoms of mental distress and well-being, disability, and self-confidence. These outcomes were measured with four questionnaires available in the participants’ native language obtained at baseline and at the end of program; (1) the Harvard Trauma Questionnaire (HTQ) [34], the World Health Organization—Five Well-Being Index (WHO-5), version 2.0, [35], the General Self-Efficacy Scale (GSE) [36], and World Health Organization Disability Assessment Schedule, version 2.0, 12-item (WHODAS) [35]. The HTQ contains 16 questions concerning symptoms related to PTSD and uses a scale ranging from 1–4 (4 = *extremely*). The WHO-5 consists of 5 questions related to well-being scored on a scale ranging from 1–5 (5 = *all the time*). The GSE aims at capturing self-beliefs used to cope with difficult demands in life and includes 10 questions scored on a scale from 1–4 (4 = *exactly true*). Where the other questionnaires are self-administered, WHODAS is an interviewer-administered questionnaire that measures level of disability with 12 items rated on a 5-point scale (5 = *cannot do*). 

### 2.5. Statistics

The data were analyzed with MATLAB (Statistics Toolbox Release 2018a, The MathWorks, Inc., Natick, MA, USA). McNemar chi-square significance test with Yates’s correction was used to test the paired categorical primary outcome data entered in a 2-by-2 contingency table. Paired *t*-tests were used to test the four secondary outcome datasets. All data sets were tested for the assumption of normal distribution with the Kolmogorov–Smirnov test [37] and for outliers using Grubbs’s test [38]. The family-wise error rate associated with four comparisons was corrected with the Bonferroni–Holms method where the *p*-value is consecutively adjusted with each test [39]. We used a two-sided *p*-value, and the threshold for significance was set to α = 0.05.

## 3. Results

### 3.1. Affiliation to Labor Market

Of the 28 participants, 15 increased their affiliation score following the program, among which 6 obtained employment. Meanwhile, 2 participants decreased their affiliation score. The McNemar analysis of the one-year incidence categorical variable (Yes/No) dichotomized from the affiliation score and measured pre-program initiation and post-program completion revealed a significant increase in affiliation (chi^2^ (1, n = 28) = 5.78, *p* = 0.016). There were unfortunately no data collected on affiliation to labor market among the 10 dropouts. The distribution of affiliation activities before and after the program is shown in Table 3.

### 3.2. Mental Well Being

There were only 16 participants with complete WHODAS, GSE, WHO-5 data sets and 15 participants with complete HTQ data set. Between outcome measures were different participants with incomplete data sets. There were no outliers, and all data sets were normally distributed. Table 4 presents the scores on the four outcomes and test results from sampled *t*-tests comparing scores at baseline and end of program for participants with complete data sets. We saw no effect of the program on GSE or WHODAS scores but did see a significant decrease in HTQ score (*p* = 0.039) and increase in the WHO-5 well-being index (*p* = 0.002). The change in the WHO-5 well-being index was also statistically significant after Bonferroni–Holms correction (*p* = 0.007).

## 4. Discussion

In this small case series study, we found an association between a 30-week program combining nature-based therapy with traditional job training and improved job market affiliation for newly arrived trauma-affected refugees with poor Danish language skills and a high psychiatric symptom load. We also found a small improvement in self-reported symptoms of PTSD and functioning in everyday situations.

### 4.1. Changes in Job Market Affiliation

The results suggest that the positive effect of horticulture and nature-based therapy on the ability to restore fatigued cognitive resources [40] and mental well-being in stress-related disorders [17] might also play a positive role in bringing refugees closer to the labor market. To understand this connection, it is valuable to draw on the results from a qualitative component of the study that we reported elsewhere [41]. Here, an interpretative phenomenological analysis of in-depth interviews performed on a subset of the participants revealed that the nature-related activities generated a feeling of competence and confidence in the ability to learn new skills. Moreover, the program fueled a feeling of connectedness and, for many, a belief in the possibility of acquiring a job or education. Since such experiences, under the headings of ‘doing’, ‘being’, and ‘belonging’ elsewhere have been suggested to comprise dimensions of meaning in the occupations of daily life [42], one might speculate if the program lay ground for a frame of mind pertinent to the labor market. 

The relevance of obtaining cultural competence and developing a sense of belonging have recently been supported in a qualitative study on refugee health professionals in Germany [43]. Despite formal records of adequate educational level and experience, this study showed that inadequate work culture knowledge, emotional challenges, and challenges with team members constituted barriers in labor market integration. Thus, while horticultural activities and vocational training at a first glance can seem misplaced for newly arrived refugees with long educations or previous high paid/status jobs, they may nurture important aspects of labor market integration if situated in the right context. For other refugees, these activities can help develop specific workplace competencies which are pivotal in civic integration and employment programs [44,45].

Moreover, from a perspective of salutogenesis, the nature environment, the composition of the program, and the activities may jointly have contributed to a sense of coherence rendering some aspect of life more comprehensible, meaningful, and manageable [46]. In accord with other empirical research aimed at strengthening sense of coherence [47,48,49], this may have mobilized mental resources and helped participants to engage in dialogues with the social workers and profit from the labor market activities that formed part of the last part of the program. 

### 4.2. Effect on Self-Reported Symptoms of PTSD and Functioning in Everyday Situations

The observed small, but statistically significant, decrease in PTSD-symptomatology and disability measured with the HTQ and WHODAS are in line with some qualitative research on war-veterans showing an effect of nature-therapy on self-assessed improvement in PTSD symptoms, and ability to participate in social activities [18,19]. It also agrees with studies linking horticulture interventions with increased mental well-being among refugees [50,51]. The observed effect is particularly noteworthy considering the high rate of psychiatric illness and symptom load among participants which were similar to what is seen in trauma-affected treatment-seeking refugee populations [52]. 

In addition to a general restorative effect of horticulture and nature-based therapy on attention and wellbeing among stress-related illnesses [17,23], the results may also be explained by the informal therapeutic conversations. Since these took place in outdoor spaces they allowed for natural pauses and reflections which in some studies have shown to enrich the therapeutic encounter [53,54], possibly by way of increased embodied awareness [55]. Considering that yoga has previously been shown to have an effect on some aspects of PTSD [56,57], the inclusion of elements of yoga in the relaxation techniques and physical exercises may also explain some of the observed effect. However, the relationship between yoga and stress-related illnesses and PTSD is still unclear as results from studies have been mixed [58]. 

While more research is needed to disentangle how the specific activities contributed to the overall effect, we believe programs guided by the following headings are of value in bringing trauma-affected refugees a step closer to the labor market: (1) a village community set in natural surroundings, (2) exercises aimed at supporting a positive body–mind experience, and (3) horticultural activities.

### 4.3. Methodological Considerations and Limitations

Several limitations are important to highlight. The absence of a randomly selected control group increases the likelihood that the observed effects were driven by other, possibly unknown, factors including placebo effect, regression toward the mean, the natural course of mental health, and investigator biases [59]. Additionally, as an uncontrolled study, it is subject to the “Hawthorne” effect, which refers to the tendency of participants to work harder and perform better merely because they are participants in a study [60]. Since we failed to collect data from dropouts we were not able to estimate the possible contribution of these factors, which further emphasizes the preliminary nature of the results. Moreover, the small sample size increased the risk of both false positive and false negative results. 

## 5. Conclusions

We found that nature-based therapy and horticulture mixed with traditional job market interventions can increase job market affiliation and mental well-being among trauma-affected newly arrived refugees. Considering a high unemployment rate and mental health problems among refugees in addition to the absence of successful Danish labor market integration programs, these preliminary results could become of crucial importance. However, considering the methodological limitations, the results primarily contribute to a strengthening of the hypothesis that environmental therapy and horticulture can have positive effects on employment rates and mental health among newly arrived refugees, and call for methodologically stronger studies to investigate the validity of these findings.

## Figures and Tables

**Figure 1 ijerph-19-04850-f001:**
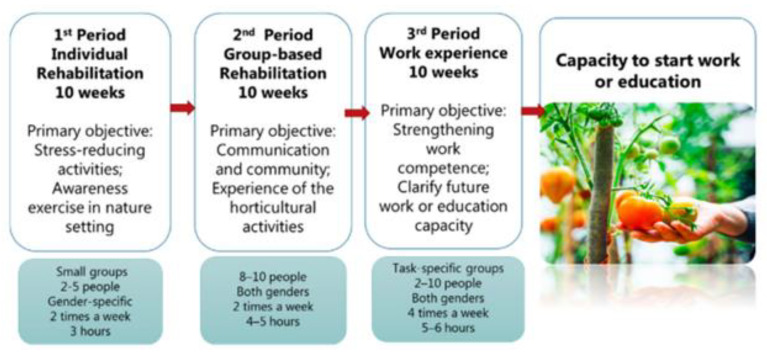
A nature-based therapy and horticulture program aimed at increasing affiliation to the labor market and improve mental well-being.

**Table 1 ijerph-19-04850-t001:** Exercises aimed at reducing psychological stress, increasing mental awareness and physical wellbeing.

**Association Exercises**
A basket with items related to nature (e.g., a rock, a wood branch, a flower) were passed around in groups of 5. After choosing and inspecting an item, participants were encouraged to describe their state of mind and what the item might symbolize in their life. The exercise was performed daily in Period 1 for approximately 20 min.
**Relaxation Techniques**
Relaxation techniques aimed at increasing bodily awareness, being present in the moment, and relieving stress and rumination were taught daily for 5–15 min. Among other things, participants were taught how to perform long exhalations and encouraged to use the techniques in situations of mental distress, e.g., during an anxiety attack.
**Physical Exercises**
Participants were daily instructed in physical exercises for 5–10 min inspired by traditional yoga. The exercises aimed at relieving stress and generating positive bodily experiences.
**Sensory Experiences in Nature**
Participants were trained in exercises aimed at increasing sensory awareness in nature. The exercises were inspired by mindfulness techniques and taught how to shift mental focus from stress, pain, and worries to present sensory experiences and the accompanying states of bodily awareness. The participants were encouraged to use exercises when experiencing emotional stress and physical pain.

**Table 2 ijerph-19-04850-t002:** Demographic and clinical characteristics of groups 1 and 2 (n = 28).

	**Mean (SD)**
Age, years	46.7 (8.38)
Time in Denmark, years	1.9 (1.03)
Education, years	7.2 (4.24)
	**Number (%)**
Female/Male	11/17 (39/61)
Work experience from homeland,	19 (68)
Met criteria for a psychiatric diagnosis *	22 (79)
Anxiety disorders ^#^	20 (71)
Depressive disorders	18 (64)
Post-Traumatic Stress Disorder	11 (39)

* Provisional DSM-IV diagnosis based on the MINI-5 interview. The diagnosis was not confirmed by a trained clinician. ^#^ Not including PTSD. SD = standard deviation.

**Table 3 ijerph-19-04850-t003:** One-year incidence of labor market before and after intervention (n = 28) *.

	Before	After
Employment	0	6
Completed internships **^#^**	4	12
Failed internships, No	4	0
Completed educational training **^#^**, No	0	4

* Each participant can be represented in more than one cell. ^#^ Includes ongoing internship/educational training at 1-year follow-up.

**Table 4 ijerph-19-04850-t004:** Effect on mental distress and level of health, disability, and self-confidence.

	Baseline	End of Program	Paired *T*-Test Statistics
Harvard Trauma Questionnaire, mean (SD)	3.29 (0.59)	2.97 (0.51)	t_14_ = 2.27, *p* = 0.039
Well-Being Index (WHO-5), mean (SD)	19 (26.47)	26.25 (18.53)	t_15_ = 0.94, *p* = 0.362
General Self-Efficacy Scale (GSE), mean (SD)	18.19 (6.12)	19.88 (5.71)	t_15_ = 0.90, *p* = 0.377
WHO Disability Assessment Schedule 2.0 (WHODAS), mean (SD)	35.12 (9.58)	40.81 (8.67)	t_15_ = 3.82, *p* = 0.002

SD = standard deviation.

## Data Availability

The data presented in this study are available on request from the corresponding author.

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
