# Peer review of "Mixing Job Training with Nature-Based Therapy Shows Promise for Increasing Labor Market Affiliation among Newly Arrived Refugees: Results from a Danish Case Series Study"

_ijerph, 2022, doi:10.3390/ijerph19084850_

Round 1

Reviewer 1 Report

Dear authors,

I include several suggestions related to your manuscript:

  1. Abstract: Abstract is the first presentation of a research. I recommend omitting specific statistics and summarize main findings in the abstract (e.g., not including sentences like "McNemar chi-square significance test of a 2-by-2 contingency table dichotomizing (Yes/No) the 1-year incidence of job market affiliation before and after 22 program showed a significant increase (p = 0.016) in affiliation". Main idea is facilitating the reading to a broad scope of researchers and practitioners that may not be familiar with these statistics.

2. Although the experiment is deeply described, I miss the theoretical framework that justifies your research. There are some references in the introduction. However, the introduction is quite short and there is not a specific section for theory.

3. I would recommend moving the "3.1 participants" description from results to the previous 2.3 subsection.

4. Main limitation, from my point of view, of the research is the number of participants. Furthermore, there were only 16 participants with complete WHODAS, GSE, WHO-5 data sets 243 and 15 participants with complete HTQ data set. So we are working with very small samples.

Author Response

Please use attached docx 

Reviewer 1

Dear authors,

I include several suggestions related to your manuscript:

  1. Abstract: Abstract is the first presentation of a research. I recommend omitting specific statistics and summarize main findings in the abstract (e.g., not including sentences like "McNemar chi-square significance test of a 2-by-2 contingency table dichotomizing (Yes/No) the 1-year incidence of job market affiliation before and after 22 program showed a significant increase (p = 0.016) in affiliation". Main idea is facilitating the reading to a broad scope of researchers and practitioners that may not be familiar with these statistics.

Response 1. We thank the reviewer for this point and recognize that omitting this information increase readability. The abstract has been revised accordingly.

Revised text (abstract)

The unemployment rate among newly arrived refugees in European countries is high and many experience mental health problems with negative consequences on integration and mental well-being. In this case-series study we investigated the effect of a 30 weeks program that included horticulture activities, nature-based therapy and traditional job training on job market affiliation and mental well-being. Fifty-two refugees met initial screening criteria and 28 met all inclusion criteria and were enrolled. The program took place in a small community and consisted of informal therapeutic conversations, exercises aimed at reducing psychological stress, increase mental awareness and physical wellbeing. At the end of the program traditional job market activities were led by social workers. Provisionary psychiatric interviews showed that at baseline 79 % met criteria for either an anxiety, depression, or PTSD diagnosis. After the program, statistical analyses revealed an increase in the 1-year incidence of job market affiliation (n = 28) and an increase in mental health according to two of four questionnaire measures (nrange = 15-16).The results strengthen the hypothesis that horticulture and nature-based therapy can help refugees enter the job market. However, the small sample size emphasizes the need for methodologically stronger studies to corroborate these preliminary findings.

  1. Although the experiment is deeply described, I miss the theoretical framework that justifies your research. There are some references in the introduction. However, the introduction is quite short and there is not a specific section for theory.

Response 2. We thank the reviewer for pointing out the lack of introduction concerning theoretical framework. We have added the following text in the introduction section.

Revised text (introduction)

[…] Against the backdrop of other significant stressors such as pre-migratory traumatic experiences (e.g. war, torture, famine) and stressors related to the migration, unemployment is moreover a potential risk factor for mental disorders among refugees [12,13], such as posttraumatic stress disorder (PTSD) and depression [14]. While research has demonstrated the detrimental impact of both unemployment and mental distress among refugees there is a lack of studies investigating possible mitigating interventions.

Meanwhile, an increasing body of research show, that nature-based therapy can have a positive impact on stress [15–17] and mental disorders, including PTSD [18,19] and anxiety [20]. Nature-based therapy covers a range of activities that pivot on experiences in nature and facilitates treatment processes. One widespread form of delivering nature-based therapy is horticulture, of which many activities have been adopted from occupational therapy [21,22]. The dominating theory corroborating the therapeutic aspect of nature is attention restoration theory (ATR) developed by Kaplan [23]. ART suggests that nature has a restorative effect on cognitive demanding forms of attention that we employ voluntarily and with effort, such as when the attention goes ‘against the grain’. Natures restorative effect is assumed to be mediated by mental states of effortless fascination and associated positive emotional and existential states, which have been substantiated by environmental studies on stress-related conditions linking the experience of drifting clouds [24], calmness and sense of freedom [25] with improved mental health.

  1. I would recommend moving the "3.1 participants" description from results to the previous 2.3 subsection.

Response 3. We agree that this would adhere more strongly to the general format of similar articles and have made the suggested change.

  1. Main limitation, from my point of view, of the research is the number of participants. Furthermore, there were only 16 participants with complete WHODAS, GSE, WHO-5 data sets 243 and 15 participants with complete HTQ data set. So we are working with very small samples.

Response 4. We agree that this is a strong limitation. The limitation is already emphasized in the limitation and conclusion and have now further been underlined in the abstract (see above)

Reviewer 2 Report

The topic and the article are interesting. However several changes must be addressed as described below:
The abstract is quite confusing, with loose ideas, not reflecting the content of the article. Furthermore, methodological details are not expected to be presented in this section. For example, McNemar chi-square significance test of a 2-by-2 contingency table dichotomizing (Yes/No) the 1-year incidence of job market affiliation before and after 22 program showed a significant increase (p = 0.016) in affiliation. 
The introduction is very interesting and well argued, but the research gap is missing. The authors present what exists, but do not evidence what does not exist in the literature and that justifies the need for the study.
Since there is no theoretical framework chapter, I expected more from the discussion. The link with theory is vague and does not really constitute a discussion. For example, in this sentence "The results suggest that the positive effect of horticulture and nature-based therapy on the ability to restore fatigued cognitive resources [33] and mental well-being in stress-related disorders [17]," the authors sprinkle in some authors but do not exactly make a discussion. Moreover, only what validates previous studies is identified (e.g. . It also agrees with studies linking horticulture interventions with increased mental well-being among refugees [37,38].) but do not discuss where the study advances knowledge.
The lack of a research gap leads to conclusions being a simple recapitulation, with no real identification of the contributions of the study.

Author Response

Please use attached docx 

Reviewer 2

The topic and the article are interesting. However, several changes must be addressed as described below:

  1. The abstract is quite confusing, with loose ideas, not reflecting the content of the article. Furthermore, methodological details are not expected to be presented in this section. For example, McNemar chi-square significance test of a 2-by-2 contingency table dichotomizing (Yes/No) the 1-year incidence of job market affiliation before and after 22 program showed a significant increase (p = 0.016) in affiliation.

Response 1. We thank the reviewer for this point and recognize that omitting this information increase readability. The abstract has been revised accordingly. Additional changes have been made to accommodate suggestions from Reviewer 1.

Revised text (abstract)

The unemployment rate among newly arrived refugees in European countries is high and many experience mental health problems with negative consequences on integration and mental well-being. In this case-series study we investigated the effect of a 30 weeks program that included horticulture activities, nature-based therapy and traditional job training on job market affiliation and mental well-being. Fifty-two refugees met initial screening criteria and 28 met all inclusion criteria and were enrolled. The program took place in a small community and consisted of informal therapeutic conversations, exercises aimed at reducing psychological stress, increase mental awareness and physical wellbeing. At the end of the program traditional job market activities were led by social workers. Provisionary psychiatric interviews showed that at baseline 79 % met criteria for either an anxiety, depression, or PTSD diagnosis. After the program, statistical analyses revealed an increase in the 1-year incidence of job market affiliation (n = 28) and an increase in mental health according to two of four questionnaire measures (nrange = 15-16).The results strengthen the hypothesis that horticulture and nature-based therapy can help refugees enter the job market. However, the small sample size emphasizes the need for methodologically stronger studies to corroborate these preliminary findings.

  1. The introduction is very interesting and well argued, but the research gap is missing. The authors present what exists, but do not evidence what does not exist in the literature and that justifies the need for the study.

Response 2. We thank the reviewer for noticing the paucity and have added a justification. To accommodate suggestions from both Reviewers theoretical background has also been added the introduction section.

Revised text (introduction)

[…] Against the backdrop of other significant stressors such as pre-migratory traumatic experiences (e.g. war, torture, famine) and stressors related to the migration, unemployment is moreover a potential risk factor for mental disorders among refugees [12,13], such as posttraumatic stress disorder (PTSD) and depression [14]. While research has demonstrated the detrimental impact of both unemployment and mental distress among refugees there is a lack of studies investigating possible mitigating interventions.

Meanwhile, an increasing body of research show that nature-based therapy can have a positive impact on stress [15–17] and mental disorders, including PTSD [18,19] and anxiety [20]. Nature-based therapy covers a range of activities that pivot on experiences in nature and facilitates treatment processes. One widespread form of delivering nature-based therapy is horticulture, of which many activities have been adopted from occupational therapy [21,22]. The dominating theory corroborating the therapeutic aspect of nature is attention restoration theory (ATR) developed by Kaplan [23]. ART suggests that nature has a restorative effect on cognitive demanding forms of attention that we employ voluntarily and with effort, such as when the attention goes ‘against the grain’. Natures restorative effect is assumed to be mediated by mental states of effortless fascination and associated positive emotional and existential states, which have been substantiated by environmental studies on stress-related conditions linking the experience of drifting clouds [24], calmness and sense of freedom [25] with improved mental health.

  1. Since there is no theoretical framework chapter, I expected more from the discussion. The link with theory is vague and does not really constitute a discussion. For example, in this sentence "The results suggest that the positive effect of horticulture and nature-based therapy on the ability to restore fatigued cognitive resources [33] and mental well-being in stress-related disorders [17]," the authors sprinkle in some authors but do not exactly make a discussion. Moreover, only what validates previous studies is identified (e.g. . It also agrees with studies linking horticulture interventions with increased mental well-being among refugees [37,38].) but do not discuss where the study advances knowledge.The lack of a research gap leads to conclusions being a simple recapitulation, with no real identification of the contributions of the study.

Response 3. We thank the reviewer for addressing the discussion section and its current limitations. In response, we have expanded on the theoretical links and suggested how the study may advance the field.

Revised text (discussion)

The results suggests that the positive effect of horticulture and nature-based therapy on the ability to restore fatigued cognitive resources [38] and mental well-being in stress-related disorders [17], might also play a positive role in bringing refugees closer to the labor market. To understand this connection, it is valuable to draw on the results from a qualitative component of the study that we reported elsewhere [39]. Here, an interpretative phenomenological analysis of in-depth interviews performed on a subset of the participants revealed that the nature-related activities generated a feeling of competence and confidence in the ability to learn new skills. Moreover, the program fueled a feeling of connectedness and for a large part a belief in the possibility of acquiring a job or education. Since such experiences, under the headings of ‘doing’, ‘being’ and ‘belonging’, elsewhere have been suggested to comprise dimensions of meaning in the occupations of daily life [40], one might speculate if the program lay ground for a frame of mind pertinent to the labor market.

The relevance of obtaining cultural competence and developing a sense of belonging have recently been supported in a qualitative study on refugee health professionals in Germany [41]. Despite formal records of adequate educational level and experience, this study showed that inadequate work culture knowledge, emotional challenges, and challenges with team members constituted barriers in labor market integration. Thus, while horticultural activities and vocational training at a first glance can seem misplaced for newly arrived refugees with long educations or previous high paid/status jobs, they may nurture important aspects of labor market integration if situated in the right context. For other refugees, these activities can help develop specific workplace competencies which are pivotal in civic integration and employment programs [42,43].

Moreover, from a perspective of salutogenesis the nature environment, the composition of the program and the activities may jointly have contributed to a sense of coherence rendering some aspect of life more comprehensible, meaningful and manageable [44]. In accord with other empirical research aimed at strengthening sense of coherence [45–47], this may have mobilized mental resources and help participants to engage in dialogues with the social workers and profit from the labor market activities that formed part of the last part of the program.

4.2. Effect on self-reported symptoms of PTSD and functioning in everyday situations

The observed small, but statistically significant, decrease in PTSD-symptomatology and disability measured with the HTQ and WHODAS are in line with some qualitative research on war-veterans showing an effect of nature-therapy on self-assessed improvement in PTSD symptoms, and ability to participate in social activities [18,19]. It also agrees with studies linking horticulture interventions with increased mental well-being among refugees [48,49]. The observed effect is particularly noteworthy considering the high rate of psychiatric illness and symptom load among participants which were similar to what is seen in trauma-affected treatment-seeking refugee populations [50].

In addition to a general restorative effect of horticulture and nature-based therapy on attention and wellbeing among stress-related illnesses [17,23], the results may also be explained by the informal therapeutic conversations. Since these took place in outdoor spaces they allowed for natural pauses and reflections which in some studies have shown to enrich the therapeutic encounter [51,52], possibly by way of increased embodied awareness [53]. Considering that yoga previous have shown to have effect on some aspect of PTSD [54,55], the inclusion of elements of yoga in the relaxation techniques and physical exercises may also explain some of the observed effect. Preliminary. However, the relationship between yoga and stress-related illnesses and PTSD is still unclear as results from studies have been mixed [56].

While more research is needed to disentangle how the specific activities contributed to the overall effect, we believe the that programs guided by the following headings are of value in bringing trauma-affected refugees a step closer to the labor market; 1) a village community set in natural surroundings, 2) exercises aimed at supporting a positive body-mind experience, and 3) horticultural activities.

Round 2

Reviewer 1 Report

I am very sorry but I think the research does not face its main problem; the sample size. Honestly, I think that 15 observations are not enough to support conclusions.

Therefore, I recommend not publishing the paper in its current version.

Author Response

Dear reviewer, 

Thank you for your succint comment. We fully agree that the sample sizes for both the main  (n = 28) and secondary (n = 15- 16) analyses are in the lower end. However, while the risk for type 1 and 2 error associated with interpretation consequently are increased, all data sets were normally distributed with no outliers. This information have now been added (see below). According to standard text books on statistics, the sample sizes were therefore "large enough". The statistical uncertainty related to the small sample is expressed in Abstract ("However, the small sample size emphasizes the need for methodologically stronger studies to corroborate these preliminary findings."), Limitation ("the small sample size increased the risk of both false positive and false negative results") and Conclusion ("considering the methodological limitations the results primarily contribute to a strengthening of the hypothesis...calls for methodologically stronger studies to investigate the validity of these findings."). On these grounds, we dont believe, in accord with Reviewer 2, that the small sample sizes should prevent publication. 

Revised text (Statistics)

2.5. Statistics

The data were analysed with MATLAB (Statistics Toolbox Release 2018a, The MathWorks, Inc., Natick, Massachusetts, United States). McNemar chi-square significance test with Yates'es correction was used to test the paired categorical primary outcome data entered in a 2-by-2 contingency table. Paired t-tests were used to test the four secondary outcome datasets. All data sets were tested for the assumption of normal distribution with the Kolmogorov-Smirnov test [37] and for outliers using Grubbs’s test [38]. The family-wise error rate associated with four comparisons was corrected with the Bonferroni-Holms method where the p-value is consecutively adjusted with each test [39]. We used a two-sided p-value, and the threshold for significance was set to α = 0.05.

Revised text (Results)

There were only 16 participants with complete WHODAS, GSE, WHO-5 data sets and 15 participants with complete HTQ data set. Between outcome measures it were different participants with incomplete data sets. There were no outliers and all data sets were normally distributed. Table 4 presents the scores on the four outcomes and test results from sampled t-tests comparing scores at baseline and end of program for participants with complete data sets. We saw no effect of the program on GSE or WHODAS scores but did see a significant decrease in HTQ score (p = 0.039) and increase in the WHO-5 well-being index (p = 0.002). The change in the WHO-5 well-being index was also statistically significant after Bonferroni-Holms correction (p = 0.007).

Reviewer 2 Report

Congrats for the revision.

Author Response

Thank you